# An Example of Personalized Treatment in HR+ HER2+ Long Survivor Breast Cancer Patient (Case Report)

Martina Panebianco [1], Beatrice Taurelli Salimbeni [1], Michela Roberto [1,2,*] and Paolo Marchetti [1,3]

1 Department of Clinical and Molecular Medicine, Oncology Unit, Sant' Andrea Hospital, University "La Sapienza", 00167 Rome, Italy; martina.panebianco@uniroma1.it (M.P.); beatrice.taurellisalimbeni@uniroma1.it (B.T.S.); paolo.marchetti@uniroma1.it (P.M.)

2 Department of Medical-Surgical Sciences and Translation Medicine, Sant' Andrea Hospital, University "La Sapienza", 00167 Rome, Italy

3 Oncology Unit, Istituto Dermopatico Dell'Immacolata—Istituti di Ricovero e Cura a Carattere Scientifico of Rome, 00167 Rome, Italy

* Correspondence: michela.roberto@uniroma1.it

**Abstract:** Background. Personalized therapy is becoming increasingly popular in oncological scenarios, not only based on molecular pharmacological targets, but also preventing any drug–drug–gene interaction (DDGI), which could lead to severe toxicities. Single nucleotide polymorphisms (SNPs), the individual germline sequence variations in genes involved in drug metabolism, are correlated to interindividual response to drugs and explain both efficacy and toxicity profiles reported by patients. Case presentation. We present the case of a woman suffering from triple-positive breast cancer; she had early-stage disease at the onset and after four years developed metastatic disease. During her history, she presented different toxicities due to antineoplastic treatments. Particularly, hypertransaminasemia was found during every line of treatment. Nevertheless, we were able to guarantee the patient an excellent therapeutic adhesion thanks to the supportive treatments and the reduction of drug dosage. Moreover, we conducted a simultaneous analysis of the patient's biochemical and genomic data thanks to Drug-PIN software, and we found several significant SNPs of the main enzymes and transporters involved in drug metabolism. Conclusion. Our case report demonstrated the relevance of DDGI in clinical practice management of a patient treated for advanced breast cancer, suggesting the role of Drug-PIN software as an easy-to-use tool to prevent adverse events during cancer treatment and to help physicians in therapeutic algorithms. However, further studies are needed to confirm these results.

**Keywords:** hormone receptor positive (HR+); HER2+ breast cancer; drug-drug-gene interaction (DDGI); single nucleotide polymorphisms (SNPs)





## 1. Introduction

The idea of personalized therapy is becoming increasingly popular, not only on the basis of molecular targeted therapy, but also by considering any potential drug–drug–gene interaction (DDGI) to prevent adverse events during cancer treatment, in order to improve patients outcome [1]. Based on this consideration, the pharmacogenomics profile to define the DDGI and the use of smart software to easily understand the risk of toxicity during treatment could be very useful in personalized cancer management.

We present the case of a woman suffering from hormone receptor-positive (HR+) and HER2+ breast cancer, with metastatic disease after four years from cancer diagnosis. During her history, she presented different toxicities due to antineoplastic treatments. Particularly, severe hypertransaminasemia was found during every line of treatment. Thus, we retrospectively analyzed the SNPs of main enzymes and transporters involved in drug metabolism and showed the utility of Drug-PIN software in the management and prevention of adverse events during chemotherapy treatment.

## 2. Case Presentation

A 36-year-old woman presented at our hospital with a palpable mass at the external left breast quadrants in December 2011. The patient underwent breast ultrasound and breast magnetic resonance imaging (MRI) that confirmed the presence of a nodular formation of about 5 cm. In addition, at least three lymphadenopathies with eccentric hilum were identified in the left axillary cable, and remote staging tests were negative. In January 2012, breast biopsy diagnosed an infiltrating ductal carcinoma of the breast, grade 3, estrogen receptor (ER)-positive (90%), progesterone receptor (PR)-positive (36%), HER2-positive (3+ by IHC), and MIB-1 36%. The pathological nature of lymphadenopathies was confirmed by fine-needle aspiration.

From January 2012 to April 2012, the patient received neoadjuvant chemotherapy with docetaxel (100 mg/m$^2$) and trastuzumab (8 mg/kg loading dose, followed by 6 mg/kg) every three weeks for four cycles followed by cyclophosphamide (600 mg/m$^2$) and doxorubicin (60 mg/m$^2$) every three weeks for four cycles. The second cycle of chemotherapy was delayed due to hypertransaminasemia grade 2 (according to common terminology criteria for adverse events (CTCAE) version 5.0), treated with glutathione and prednisone 25 mg/die for five days. Treatment with docetaxel was resumed at 25% reduced dose, reporting only grade 1 hypertransaminasemia during subsequent cycles. In August 2012, the patient underwent bilateral mastectomy and axillary node resection with a partial response to neoadjuvant chemotherapy (ypT1c ypN1 Mx).

From October 2012 to April 2013, the patient continued adjuvant therapy with trastuzumab administered subcutaneously (600 mg) every three weeks to complete one year of treatment and hormone therapy with tamoxifene plus luteinizing hormone-releasing hormone (LHRH) inhibitor since September 2012.

In January 2014, after the finding of thickening of the subcutaneous areola, the patient underwent surgical radical removal of the nipple and left areola. The diagnosis was intraductal carcinoma of cribriform type in galactophore ducts of high-grade sec. WHO 2012 and Paget's disease, ER 90% PgR negative.

She continued regular clinico-radiological follow-up, negative for locoregional and distance recurrence, until January 2016 when a total body contrast-enhanced computed tomography (CECT) scan and an 18-fluor-fluorodeoxyglucose positron emission tomography-computed tomography (18F-FDG PET/CT) confirmed the presence of bilateral pulmonary, mediastinal pathological lymph nodes and bone metastases. According to the disease stage, biomolecular tumor characteristics, and clinical conditions (PS 0 sec. ECOG), in January 2016, the patient started first-line therapy with pertuzumab plus trastuzumab and docetaxel every three weeks at standard doses. She reassumed the same regimen up to four courses and then she continued with dual HER2-blockage for a further three cycles. The CECT scan after three courses of treatment showed a partial response, but after the second cycle the patient showed grade 2 mucositis and grade 3 hypertransaminasemia, treated with glutathione and prednisone 25 mg/die for five days, and the third cycle was delayed. Antigens of hepatitis were tested and resulted negative.

In August 2016, an 18F-FDG PET/CT revealed lymph node and bone progression disease. Thus, she started second-line therapy with T-DM1 at 3.6 mg/kg every three weeks with concurrent denosumab every four weeks. From August 2016 to April 2018, she received 28 cycles. After the ninth cycle, due to the occurrence of grade 2 hypertransaminasemia, glutathione was introduced in chemotherapy regimens with the resolution of toxicity.

In April 2018, due to a lymph node and lung disease progression, the patient was enrolled in the phase III clinical trial SOPHIA (NCT02492711) and randomized to the control arm. The screening laboratory findings showed a grade 3 hypertransaminasemia that was treated with glutathione infusion. At the resolution of toxicity, since the patient had been randomized into the control arm, she began treatment with trastuzumab (8 mg/kg loading dose, followed by 6 mg/kg) and navelbine (25 mg/mq d1–8 every three weeks). The treatment was poorly tolerated with grade 4 neutropenia and grade 2 anemia after

the first cycle and grade 4 neutropenia after the second cycle with 50% dose reduction and secondary prophylaxis with granulocyte colony-stimulating factor (G-CSF). Due to hematological toxicities, she continued treatment without navelbine, and she received four courses with trastuzumab.

In October 2018, a further lymph node and lung disease progression was detected, and then she started fourth line with capecitabine (1000 mg/mq bid d1–14) plus lapatinib (1250 mg once daily) every three weeks. The patient received IX cycle from November 2018 to June 2019. After the first cycle, she presented grade 1 hypertransaminasemia, which was resolved with glutathione infusion.

After a further lymph node and lung disease progression, she received fifth-line therapy with nab-paclitaxel 260 mg/m$^2$ and trastuzumab every three weeks with primary prophylaxis with G-CSF. She received nine courses from July 2019 to December 2019, and she presented only grade 1 hypertransaminasemia.

Due to the evidence of complete response and the patient's preference to discontinue alopecitizing chemotherapy, she continued treatment with trastuzumab alone until March 2020, when an 18F-FDG PET/CT revealed lymph node and lung relapse. Then, in April 2020, she started sixth-line therapy with gemcitabine 1000 mg/m$^2$·d1, d8 plus trastuzumab 6 mg/kg every three weeks. After the first cycle, she presented grade 2 hypertransaminasemia and grade 2 thrombocytopenia, so she stopped the treatment; due to the occurrence of pain in the left arm (VAS 8), we prescribed oxycodone plus naloxone. At the fourth cycle, she showed grade 3 hypertransaminasemia, and we decided to reduce gemcitabine dose, but the toxicity persisted. Therefore, liver virology was tested again. Moreover, at this time point (Figure 1), since the new technology for DDGI had become available at our center, we evaluated her genomic polymorphisms, including genes encoding for the main drug metabolism enzymes, and simultaneously analyzed all her clinical, biochemical, and genomic data by using the new Drug-PIN system comprehensive approach (https://aousa.drug-pin.com/app/ (accessed on 31 March 2020)) In particular, for the SNP analysis, the patient's DNA was extracted from samples of 5 mL of peripheral blood, using the automatic QIAsymphony system for the extraction of nucleic acid (Qiagen, Hilden, Germany), and then the latter was processed using a next-generation sequencing platform Ion Chef/Ion S5 system (Thermo Fisher Scientific, Waltham, MA, USA) according to the manufacturer's instructions.

Due to the results of the analysis (Table S1), we decided to maintain the reduced dose of gemcitabine and to introduce glutathione as part of cancer treatment from the sixth cycle of chemotherapy.

From August to December 2020, the patient received 25% reduced dose gemcitabine with four vials of glutathione, and she showed maximum grade 2 liver toxicity.

In December 2020, an 18F-FDG PET/CT evaluation revealed oligometastatic lymph node and bone progression; therefore, we decided to treat with local radiotherapy on sternal dumbbell and left later-cervical, supraclavicular, and retro-pectoral lymph nodes, maintaining the same chemotherapy regimen. Today, the treatment with gemcitabine and trastuzumab is still ongoing, and the patient has reported an overall survival (OS) of 111 months from first diagnosis and of 50 months from recurrence.

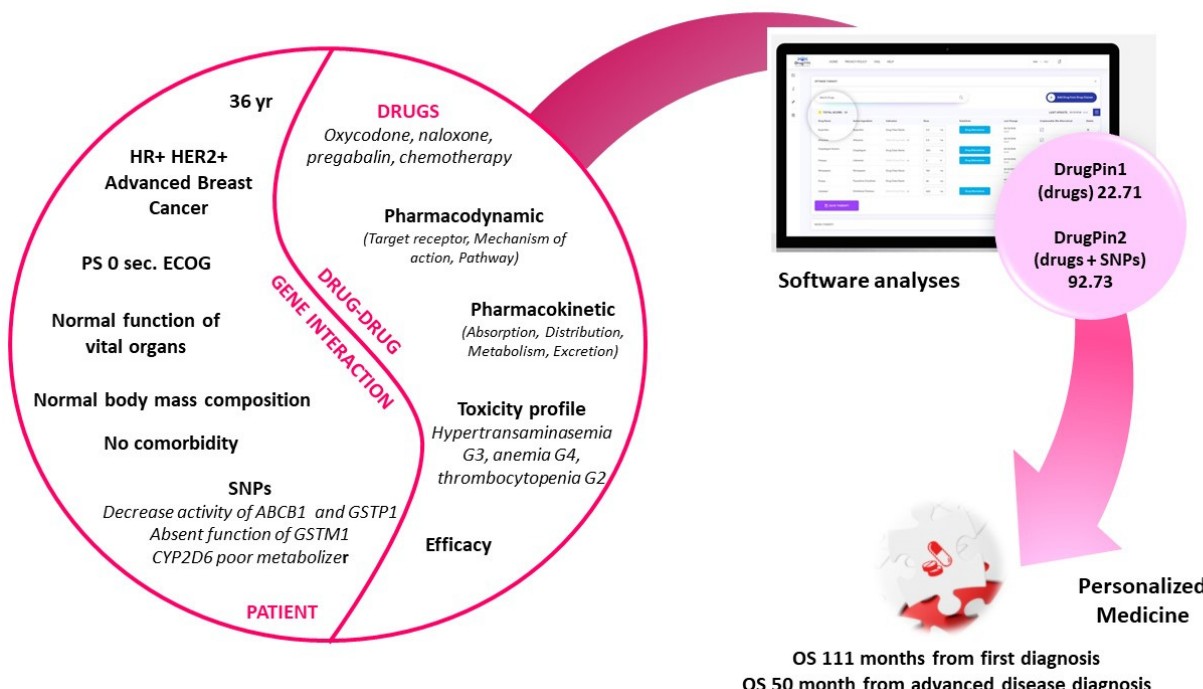

**Figure 1.** Personalized medicine thanks to new Drug-PIN system comprehensive approach. PS, Performance Status; ECOG, Eastern Cooperative Oncology Group; OS, Overal Survival; SNP, Single Nucleotide polymorphism.

## 3. Discussion

The idea of personalized therapy is becoming increasingly popular, not only based on molecular targets, but also preventing any drug–drug–gene interactions (DDGIs), which could lead to serious adverse reactions, especially in patients who receive polypharmacotherapy due to comorbidities [1].

The pharmacogenomics profiling, including all cytochromes P450 (CYPs) enzyme superfamily, P-glycoprotein, ATP-binding cassette transporters, and detoxifying and DNA-repairing enzymes [2,3], is of fundamental importance to define DDGIs, which result from the combination between drug–drug interaction (DDI) and individual genetic polymorphisms [4]. In fact, single nucleotide polymorphisms (SNPs), the individual germline sequence variations in these genes, are involved in interindividual response to drugs and explain both efficacy and toxicity profiles reported by patients [5]. Not all the SNPs are "functional", because only 1% of them are located in the coding portions of the DNA. To date, approximately 50,000–250,000 functional SNPs conferring a potential biological effect have been identified [6].

Two major examples in order to reflect the importance of SNPs in clinical practice concern those point mutations affecting dihydropyrimidine dehydrogenase (DPD, encoded by the DPYD gene), methylene tetrahydrofolate reductase (MTHFR), and thymidylate synthase (TYMS) implicated in fluoropyrimidine metabolism, as well as the SNPs of uridine diphosphate glucuronosyltransferases (UGTs) and CYP3A4*1B and CYP3A5*3 enzymes, implicated in irinotecan metabolism. In particular, based on UGT1A1 and DPYD polymorphisms, drug dose modification was recommended in clinical practice. Moreover, excision repair 1 endonuclease noncatalytic subunit (ERCC1), X-ray repair cross-complementing protein 1 (XRCC1), and glutathione S-transferase P1 (GSTP1) gene variants potentially impact the outcome of platinum derivative therapies, whereas ATP-binding cassette subfamily B member 1 (ABCB1) and CYP3A4*1B and CYP3A5*3 variants were correlated to the enzyme activity and outcome of taxanes [7,8].

Several studies have evaluated correlations between toxicities and therapeutic outcomes in patients with breast cancer in the context of hormone therapy, chemotherapy, and radiotherapy [9–15].

We presented the case of a patient suffering from high-grade HR+ HER2+ cancer who received six treatment lines for the metastatic disease. As reported in Figure 2, during the different treatment lines, a series of toxicities were found, particularly grade 3 hypertransaminasemia and grade 2 mucositis in the course of first-line treatment with pertuzumab-trastuzumab plus docetaxel and grade 2 hypertransaminasemia under neoadjuvant treatment with docetaxel–trastuzumab, second-line treatment with trastuzumab–emtansine, and sixth-line treatment with gemcitabine–trastuzumab. Moreover, the patient presented hematological toxicity with grade 4 neutropenia and grade 2 anemia in the course of third-line treatment with vinorelbine–trastuzumab and grade 2 thrombocytopenia under sixth-line treatment with gemcitabine–trastuzumab.

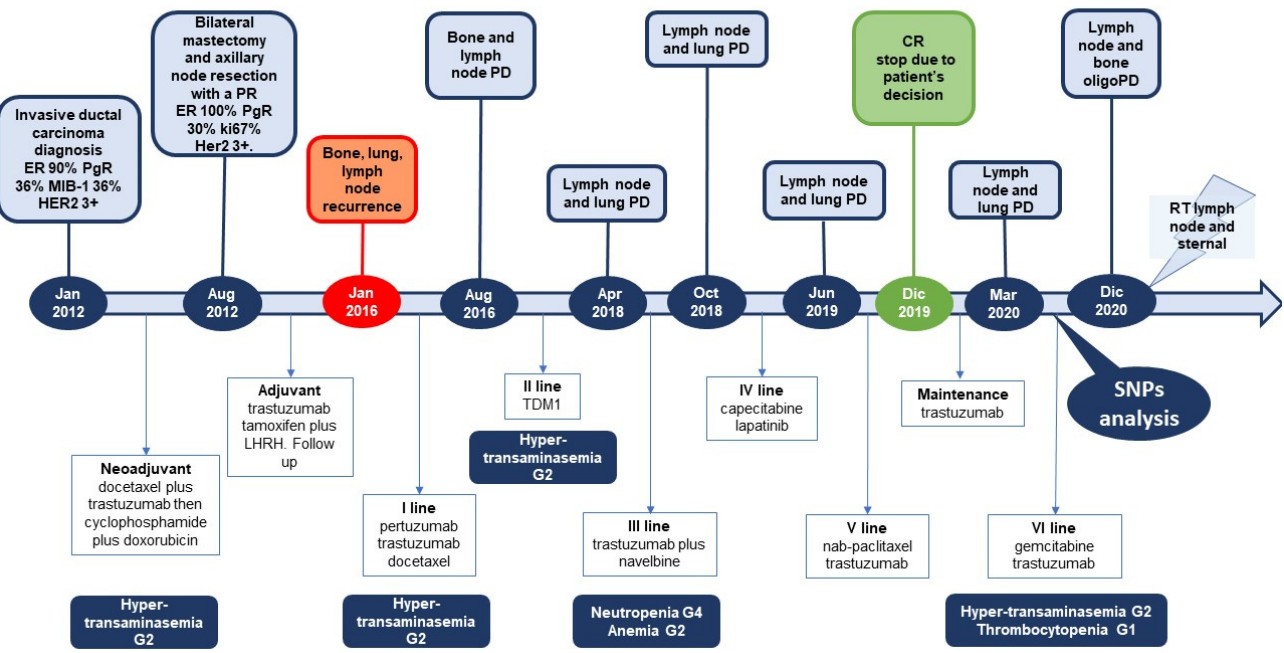

**Figure 2.** Timeline of patient's clinical history and toxicity reported in treatment lines. Legend: ER, estrogen receptor; PgR, progesterone receptor; HER2, human epidermal growth factor receptor 2; LHRH, luteinizing hormone-releasing hormone; PD, progression disease; CR, complete response; RT, radiotherapy; SNPs, single nucleotide polymorphisms.

Considering the adverse events reported and the availability of DDGI analysis at our center, we conducted the analysis of 132 SNPs (Table S1), including the main SNPs involved in drug transporters and metabolism, oxidation–reduction, DNA repair, and lipid metabolism and coagulation, as well as different receptors.

Circulating taxanes are transported to the hepatocytes by solute carrier organic anion transporter family member 1B3 (SLCO1B3) and are converted into their metabolites by CYP450 enzymes. Finally, ATP-binding cassette transporters ABCB1 and ABCC2 eliminate the metabolites through bile canaliculi. Polymorphisms of genes encoding for proteins involved in the transport and clearance of taxanes reduce excretion of the drugs, leading to the development of toxicity in patients [16]. In particular, as reported in Table S1, three heterozygous polymorphisms of ABCB1 (ABCB1-C1236T, ABCB1-C3435T, and Intron Variant rs4148738) and a homozygous SNP of ABCC2-C24T were identified in our case. It was evidenced that patients with the "CT/TT" genotype of ABCB1 (C1236T) gene showed better tumor response to docetaxel therapy than those with "CC" in course of neoadjuvant chemotherapy for locally advanced breast cancer [17], but it was associated with higher neurotoxicity in non-small-cell lung cancer (NSCLC) treated with taxane-based regime. In fact, the area under the concentration–time curve (AUC) of patients treated with docetaxel increased for ABCB1 mutation subgroups [18]. In our case report, a grade 2 liver toxicity

was evidenced under all taxane-based therapy, and we obtained a partial response to neoadjuvant chemotherapy as well metastatic setting taxanes.

In addition, the patient also presented a reduced activity of oxidation–reduction enzymes, involved in the prevention of cell damage by free radicals. In fact, the presence of heterozygous SNP of glutathione S-transferase protein 1 (GSTP1) A313G and GSTM1 deletion lead to decreased activity enzyme of GSTP1 and absent function of GSTM1, respectively (Table S1). However, the effect that GST polymorphisms might have on toxicity to taxanes is not well understood, but some studies have linked GSTP1 polymorphisms and hematological and neurological toxicity to taxanes [19–23]. In our case, the prevalent toxicity evidenced in course of taxane therapy was hepatological.

As regards the reported liver toxicity, it is also necessary to evaluate the concomitant therapy consisting of morphine derivatives (oxycodone predominantly) that the patient took for antalgic purposes. In fact, oxycodone is extensively metabolized in the liver via CYP3A4/5 to noroxycodone (45%) and via CYP2D6 to oxymorphone (19%) [24]. Allelic variants altering CYP2D6-mediated metabolism could be associated with reduced efficacy of hydrocodone and increased toxicity of codeine, each of which relies entirely on the CYP2D6 enzyme for phase 1 metabolism. In one study, such alterations were not accompanied by increased adverse events [25]. However, individual cases of reduced oxycodone efficacy [26] or increased toxicity [27] in CYP2D6 poor metabolizers have been reported [28]. As reported in Table S1, our patient was a carrier of CYP2D6 poor metabolizer.

The severe hematological toxicity (neutropenia grade 4 and anemia grade 2) that the patient developed during treatment with vinorelbine is also interesting, but there was no evidence of association between vinorelbine clearance and CYP3A5*1/*3 genotype or the ABCB1 SNPs tested for [29,30].

Moreover, we conducted an analysis through Drug-PIN software that integrates the information derived from patient SNPs with concomitant drugs and chemotherapy assumed. The Drug-PIN technology can be defined as a "Functional Biochemistry based on Genomic Profile" (http://www.drug-pin.com/index.html (accessed on 31 March 2020)). The result of the DDGI analysis carried out by the Drug-PIN software is represented by a numerical score to be understood in a penalizing sense: the greater the associated number, the more dangerous the drug cocktail will be. In particular, we calculated two metabolizer scores: (i) DrugPin1 included chemotherapy (taxanes) and concomitant medications (pregabalin, oxycodone, and naloxone hydrochloride); (ii) DrugPin2 was the global score, including the DrugPin1 and SNP profile in the software analysis. As regards our patient, DrugPin1 and DrugPin2 were 22.71 and 92.73. These findings show that it is now essential for the oncology community to evaluate patients globally, also taking into account the genomic aspect. In fact, in this case, although the DrugPin1 score showed a moderate risk of developing toxicity derived from the interaction between concomitant drugs and chemotherapy, the DrugPin2 score showed a high risk of developing adverse events when information from the genomic profile was integrated with the therapy assumed.

## 4. Conclusions

We presented the case of a patient with advanced breast cancer and a peculiar genomic profile that resulted in reduced activity of both enzymes and transporters involved in the metabolism of chemotherapeutics commonly used in clinical practice. In the course of her clinical history, the patient developed a series of toxicities, especially liver toxicities, which we tried to associate with the SNPs identified by genomic analysis. Thanks to the supportive treatments, especially in favor of hepatic detoxification, and the reduction of the dosage of drugs, we were able to guarantee the patient an excellent therapeutic adhesion. However, the possibility to use an easy-to-use pharmacogenomics tool such as Drug-PIN software [1] to prevent toxicities during cancer treatment through the evaluation of both clinical and genomic factors, as well as by taking into account the use of concomitant drugs, could ensure the application of personalized medicine and better compliance with treatments. Further studies are necessary for the identification of predictive factors of

toxicity of treatments used in order to ensure maximum benefit to patients and minimum toxicity of proposed therapies.

**Supplementary Materials:** The following are available online at https://www.mdpi.com/article/10.3390/curroncol28030184/s1, Table S1: Panel of SNPs investigated.

**Author Contributions:** Conceptualization, M.P., B.T.S. and M.R.; methodology, M.P. and B.T.S.; software, M.R.; validation, M.R. and P.M.; formal analysis, M.P.; investigation, M.P. and B.T.S.; resources, M.R and P.M.; data curation, M.R.; writing—original draft preparation, M.P. and B.T.S.; writing—review and editing, M.P., B.T.S. and M.R; visualization, M.R.; supervision, M.R.; project administration, M.R.; funding acquisition, M.P. and B.T.S. All authors have read and agreed to the published version of the manuscript.

**Funding:** This research received no external funding.

**Institutional Review Board Statement:** Not applicable.

**Informed Consent Statement:** Informed consent was obtained from all subjects involved in the study.

**Data Availability Statement:** No new data were created or analyzed in this study. Data sharing is not applicable to this article.

**Conflicts of Interest:** The authors have no conflict of interest to declare.

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
