# Peer review of "An Example of Personalized Treatment in HR+ HER2+ Long Survivor Breast Cancer Patient (Case Report)"

_curroncol, doi:10.3390/curroncol28030184_

Round 1

Reviewer 1 Report

Interesting case report with a common course for metastatic triple-positive patient. Actually, the patient is 50 months from recurrence. Today, the expected median OS for HER2-positive patients in the metastic setting is 60 months. There are some issues which should be resolved in a revised version:

-please give background data for recurrent glutathione infusions and prednisone treatment

-what is the impact of described genomic analysis for current clinical practice or future trials?

-what are the costs of described analysis?

-Did the authors changed ongoing therapies due to genomic profiles or SNP analysis?

-please provide register number for SOPHIA trial (NCT?)

-please re-check spelling carefully: softwer vs. software; eccentric ilium > eccentric hilum?; CTCAE grading: grade 2 instead of G2; HER2 vs. Her2, T-DM1 vs TDM1

-page 2, line 66: ypT1c ypN1 Mx

Author Response

Thank you for your corrections. 

-please give background data for recurrent glutathione infusions and prednisone treatment. The following studies demonstrated a protective role of glutathione in liver injury setting:

Chen Y, Dong H, Thompson DC, Shertzer HG, Nebert DW, Vasiliou V. Glutathione defense mechanism in liver injury: insights from animal models. Food Chem Toxicol. 2013 Oct;60:38-44. doi: 10.1016/j.fct.2013.07.008. Epub 2013 Jul 12. PMID: 23856494; PMCID: PMC3801188.

Eroglu N, Erduran E, Reis GP, Bahadır A. Therapeutic effect of N-acetylcysteine on chemotherapy-induced liver injury. Ir J Med Sci. 2020 Nov;189(4):1189-1194. doi: 10.1007/s11845-020-02219-1. Epub 2020 Apr 1. PMID: 32239424.

-what is the impact of described genomic analysis for current clinical practice or future trials? As reported in conclusion paragraph in lines 255-261 page 6: “However, the possibility to use an easy –to-use pharmacogenomics tool like DrugPin software [1] to prevent serious adverse reactions during cancer treatment, through the evaluation of both clinical and genomic factors, as well as by taking into account the use of concomitant drugs, could ensure the application of personalized medicine and better compliance with treatments. Further studies are necessary for the identification of predictive factors of toxicity to treatments used in order to ensure maximum benefit to patients and minimum toxicity to proposed therapies”.

-what are the costs of described analysis? The cost of genetic analysis is about 200 €. In our opinion it is not so expensive, considering the economic saving resulting from reduced hospitalization and the prevention of adverse effects In fact, although randomized clinical trials are still needed, on the basis of the score resulting from the DrugPin software, it is possible to decide before starting a treatment the reduction of drug dose or modulation of the support therapy.

-Did the authors changed ongoing therapies due to genomic profiles or SNP analysis? As reported in the “Case presentation” paragraph and in Figure 1, we performed genomic analysis on march 2020, since the new technology for DDGI became available at our centre. Therefore, our considerations and comments have been extrapolated after the analysis of polymorphisms. Therefore, the oncological treatment has not been modified based on genomic analysis.

-please provide register number for SOPHIA trial (NCT?) We added the register number for SOPHIA trial NCT in line 95 page 2: “In April 2018, due to a lymph node and lung disease progression, the patient was enrolled in the phase III clinical trial SOPHIA (NCT02492711) and randomized to the control arm.”

-please re-check spelling carefully: softwer vs. software; eccentric ilium > eccentric hilum?; CTCAE grading: grade 2 instead of G2; HER2 vs. Her2, T-DM1 vs TDM1. We corrected the text as you suggested.

-page 2, line 66: ypT1c ypN1 Mx. We corrected the text as you suggested.

Reviewer 2 Report

Abstract

Page 1 Line 24 and Line 27 – “DrugPin software”

Case Presentation

Page 2 Line 48 – “after the evidence at palpation of a” – maybe change to “with a palpable”?

Page 2 Line 97 – “had fallen into the control arm” – maybe change to “had been randomized into the control arm”

Page 3 Line 109 – “lungs” – maybe change to “lung”

Page 3 Line 116 – I think “g1, g8” is meant to be “d1, d8”

Page 3 Line 132 – “maintain reduced the dose” maybe change to “maintain the reduced dose”

Figure 2

The timeline would be better presented on an interval scale, not ordinal.

Conclusion

Page 6 Line 255 – “easy –to-use” change to “easy-to-use”

Throughout

There is some switching between “serious adverse events” and “serious advent reactions”. The authors should check at each occurrence that the appropriate term is being used.

Author Response

Abstract

Page 1 Line 24 and Line 27 – “DrugPin software”

We corrected the text as you suggested

Case Presentation

Page 2 Line 48 – “after the evidence at palpation of a” – maybe change to “with a palpable”? We corrected the text as you suggested.

Page 2 Line 97 – “had fallen into the control arm” – maybe change to “had been randomized into the control arm”. We corrected the text as you suggested.

Page 3 Line 109 – “lungs” – maybe change to “lung”. We corrected the text as you suggested.

Page 3 Line 116 – I think “g1, g8” is meant to be “d1, d8”. We corrected the text as you suggested.

Page 3 Line 132 – “maintain reduced the dose” maybe change to “maintain the reduced dose”. We corrected the text as you suggested.

Figure 2

The timeline would be better presented on an interval scale, not ordinal.

On interval scale we don’t able to represented all toxicities reported for each line treatment.

Conclusion

Page 6 Line 255 – “easy –to-use” change to “easy-to-use”. We corrected the text as you suggested.

Throughout

There is some switching between “serious adverse events” and “serious advent reactions”. The authors should check at each occurrence that the appropriate term is being used.

We checked the correct use of serious adverse events.